# Replay across Experiments:
# A Natural Extension of Off-Policy RL

**Dhruva Tirumala**[1,2] *    **Thomas Lampe**[1]    **Jose Enrique Chen**[1]    **Tuomas Haarnoja**[1]

**Sandy Huang**[1]    **Guy Lever**[1]    **Ben Moran**[1]    **Tim Hertweck**[1]    **Leonard Hasenclever**[1]

**Martin Riedmiller**[1]                **Nicolas Heess**[1]                **Markus Wulfmeier**[1]

[1]Google DeepMind
[2]University College London (UCL)

## ABSTRACT

Replaying data is a principal mechanism underlying the stability and data efficiency of off-policy reinforcement learning (RL). We present an effective yet simple framework to extend the use of replays across multiple experiments, minimally adapting the RL workflow for sizeable improvements in controller performance and research iteration times. At its core, Replay across Experiments (RaE) involves reusing experience from previous experiments to improve exploration and bootstrap learning while reducing required changes to a minimum in comparison to prior work. We empirically show benefits across a number of RL algorithms and challenging control domains spanning both locomotion and manipulation, including hard exploration tasks from egocentric vision. Through comprehensive ablations, we demonstrate robustness to the quality and amount of data available and various hyperparameter choices. Finally, we discuss how our approach can be applied more broadly across research life cycles and can increase resilience by reloading data across random seeds or hyperparameter variations.

## 1    INTRODUCTION

In the last few years, reinforcement learning (RL) has transitioned from a topic of academic study to a practical tool for the generation of controllers across various real-world applications (Degrave et al., 2022; Bellemare et al., 2020; Lazic et al., 2018; Lichtlé et al., 2022). These advancements have been enabled, in part, by recent advancements in algorithms improving data efficiency, robustness, and controller performance (Haarnoja et al., 2019; Lillicrap et al., 2016; Schwarzer et al., 2023). Nevertheless, many problems remain hard to solve with RL. For instance, high-dimensional and partial observations (e.g. egocentric camera views), high-dimensional action spaces, or reward functions that provide inadequate learning signal can all lead to poor asymptotic performance, high variance, low data efficiency, and long training times.

A major challenge in RL remains the interaction between data collection and learning. Unlike in the case of supervised learning where all data is available and static, in online RL both the quality and quantity of data available for policy or value function learning changes over the course of an experiment. The difficulty of collecting suitable data, and the complex interactions between data collection and function optimization leads to various problems including failure to learn, instabilities, and the premature convergence of function approximators early in learning (Nikishin et al., 2022; Ash & Adams, 2020).

Experience replay (Lin, 1992) has become a popular component of most, modern off-policy RL algorithms. Storing data collected over the course of an experiment and continuously training policy and value functions with this growing dataset can greatly increase data efficiency and stability of RL algorithms. It does not, however, address issues around premature convergence of the function approximators (Ash & Adams, 2020), nor does it directly provide a solution to reusing data from prior experiments. In this paper, we investigate the possibility of extending the use of replay and

---

*Correspondence can be directed to: {dhruvat, mwulfmeier}[at]google.com

reusing data in an iterative setting. Our main insight is that a minimal change to the RL workflow can greatly improve the asymptotic performance of off-policy reinforcement learning algorithms. By reusing interaction data from prior training runs it can further reduce the overall experiment time and thus speed up research iterations.

A number of studies have previously investigated how prior data (including expert data) can be used to kick-start RL training (Vecerik et al., 2017; Singh et al., 2020b; Nair et al., 2020; Walke et al., 2022; Ball et al., 2023). We find that the simplest approach, mixing prior and online data with a fixed ratio, is particularly effective across a wide range of application scenarios and algorithms. Our method dubbed Replay across Experiments (RaE) performs as well as or better than alternative approaches with fewer hyperparameter choices to be made. Furthermore, for domains that are normally hard to solve even with state-of-the-art algorithms, we find that it can be advantageous to perform multiple training iterations giving each iteration access to all data from prior training runs, effectively realising a minimalist perspective to lifelong learning. We hope that RaE's simplicity will enable straightforward integration into existing infrastructure and help improve the efficiency of RL workflows.

The main contributions of our work are as follows:

- Introduce and empirically validate our simple strategy to solve challenging tasks. Demonstrate state-of-the art performance on a number of domains including when reusing data from the publicly available offline RL Unplugged benchmark (Gulcehre et al., 2020).
- Demonstrate that our approach works across multiple algorithms including DMPO, D4PG, CRR and SAC-X.
- Compare RaE to common baselines and recent state-of-the-art methods. Highlight the combination of factors that leads to the effectiveness of our approach.
- Provide additional ablations to highlight the robustness of RaE to quantity and quality of data collected and hyper-parameters.

## 2 METHOD

### 2.1 BACKGROUND

We consider the reinforcement learning (RL) problem in which an agent observes the environment, takes an action, the environment changes its state and the agent receives a reward in response. The environment is modeled as a Markov Decision Process (MDP) consisting of the state space $S$, the action space $A$, the transition probability $p(s_{t+1}|s_t, a_t)$ and reward $r(s_t, a_t)$ when taking action $a_t$ in state $s_t$. The agent's behavior is specified using a deep neural network policy $\pi(a_t|s_t; \theta)$ with parameters $\theta$; which we will denote as $\pi(a_t|s_t)$ for brevity.

We optimize the agent to maximize the sum of discounted future rewards, as denoted by:

$$J(\pi) = \mathbb{E}_{\rho_0(s_0), p(s_{t+1}|s_t, a_t), \pi(a_t|s_t)} \Big[ \sum_{t=0}^{\infty} \gamma^t r_t \Big], \tag{1}$$

where $\gamma \in [0, 1]$ is the discount factor, $r_t = r(s_t, a_t)$ is the reward and $\rho_0(s_0)$ is the initial state distribution.

Given a policy $\pi$, the state-action value function, or critic, $Q(s_t, a_t)$ is defined as the expected discounted return when taking an action $a_t$ in state $s_t$ and then following the policy.

$$Q(s_t, a_t) = r(s_t, a_t) + \gamma \mathbb{E}_{p(s_{t+1}|s_t, a_t), \pi(a|s_t)}[Q(s_{t+1}, a)]. \tag{2}$$

Modern off-policy algorithms (Haarnoja et al., 2019; Barth-Maron et al., 2018; Abdolmaleki et al., 2018) usually operate through simultaneous or alternating optimization of policy and state-action value function. In this context, *experience replay* (Lin, 1992) is a fundamental mechanism to decouple data collection from policy and value-function optimization. Data is collected by one or multiple policies (which are commonly obtained over the course of a single experiment) and is stored in a replay buffer from which it can be retrieved to compute updates to the policy or value function. This approach has multiple benefits including data efficiency, reduced variance of the updates and smoothing the learning process (Mnih et al., 2015).

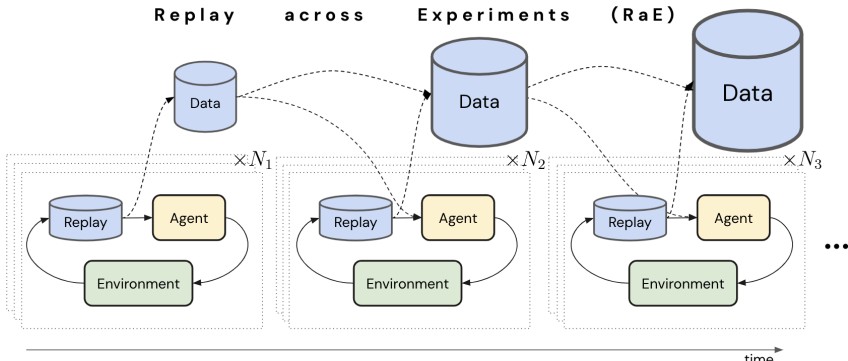

Figure 1: Intuitive schema for the use of regular replay in off-policy RL and the extended application in RaE. In every experiment, the agent can learn from a mixture of data from all previous experiments to accelerate learning and increase asymptotic performance.

The success of off-policy algorithms and the need for data-efficient and safe learning schemes has led to growing interest in approaches that allow to reuse data across experiments. One end of a spectrum aims to learn policies entirely offline from a pre-generated, fixed corpus of training data (Siegel et al., 2020; Wang et al., 2020; Kumar et al., 2020; Singh et al., 2020a; Ajay et al., 2021). This usually requires specialist algorithmic modifications to avoid instabilities when no online data is available. It is often desirable to combine prior data with new data gathered during a learning experiment. In the simplest case a policy (and value function) pretrained offline can be then finetuned online, on the same or a related task. While various offline-RL algorithms have been optimised for this setting, simpler off-policy approaches can work well with sufficiently diverse data (Yarats et al., 2022).

The second perspective to re-using data is the straightforward reloading into the replay buffer during online learning. This approach has been successful in bootstrapping learning on tasks with expert demonstrations (Vecerik et al., 2017; Nair et al., 2018; Walke et al., 2022) or to transfer to new tasks by incorporating data from scripted controllers (Singh et al., 2020b; Smith et al., 2023; Jeong et al., 2020). Finetuning offline learned agents and mixing data into a replay is further combined in the AWAC algorithm (Nair et al., 2020). It operates in two phases: in the first phase learning is performed entirely offline by pre-loading the replay buffer with offline data. Then, after a specified number of training steps, online data is mixed into the replay and learning proceeds off-policy. AWAC proposes a specific algorithm for both phases that is similar to the formulation of the offline-RL algorithm CRR (Wang et al., 2020).

While these methods have shown promising gains in data efficiency and performance, they each come with specific assumptions that increase complexity, implementation effort, and restrict their application. These include the requirement for multiple training stages (Singh et al., 2020b; Smith et al., 2023), the introduction of additional training losses (Vecerik et al., 2017; Nair et al., 2018; Song et al., 2022) or validation with specific architectures and algorithms (Nair et al., 2020; Walke et al., 2022; Ball et al., 2023) which typically require domain-specific hyperparameter tuning. The core idea that we are exploring in this paper is that the reuse of prior data can be implemented in the context of most contemporary off-policy RL workflows in a very simple way that is robust and leads to excellent results, without many of the algorithmic complexities or hyper-parameter choices of prior work.

## 2.2 REPLAY ACROSS EXPERIMENTS

The basic insight of this work is that reusing data across multiple experiments in off-policy learning is a very simple but effective way to accelerate training and improve final performance. Our key argument is the extension of this insight across all experimentation during a project. Prior data can be used to bootstrap new training runs, and it can further be beneficial to train difficult tasks in multiple iterations, bootstrapping later iterations with the data collected during earlier ones.

This idea is illustrated in Figure 1. Whereas off-policy RL is generally focused on reusing data during a single experiment, we consider sequences of experiments where data from earlier training runs

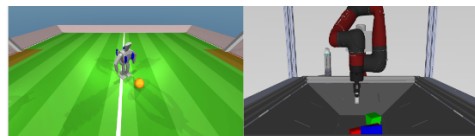 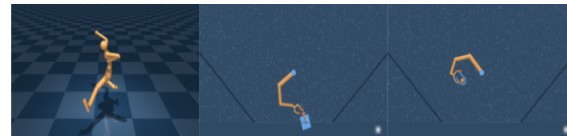

Figure 2: **Domains:** (from left to right) Locomotion, Manipulation stacking and the RL Unplugged domains - Humanoid run, Manipulator insert peg, Manipulator insert ball.

is made available during later ones[1]. At the beginning of each training run, policy and value-function are re-initialized in line with stand-alone experiments. The only required algorithmic change is the availability of a second replay mechanism that allows replaying prior and online data with a particular fixed ratio throughout the course of training (we use a naive 50/50 mix of offline and online data for our main results, without optimizing this ratio).

This simple approach performs very well without introducing additional algorithmic complexities or the need for hyper-parameter tuning. By learning in the mixed online setting, the agent controls the data distribution and in line with recent results in offline RL (Yarats et al., 2022; Lambert et al., 2022), we show that given the right data distribution, existing off-policy algorithms are sufficient for effectively using this offline data. This workflow does not require any changes with respect to the RL agent itself and is generally agnostic to agent and architecture changes across experiments. In order to emphasize this aspect, for the main results presented in Section 3.3, we integrate RaE into existing algorithmic workflows across all of the domains considered. Consequently, we evaluate RaE across a set of state-of-the-art agents including DMPO (Abdolmaleki et al., 2018), SAC-Q (Riedmiller et al., 2018), CRR (Wang et al., 2020) and D4PG (Barth-Maron et al., 2018) across a number of complex domains.

## 3 EXPERIMENTS

In this section, we evaluate the performance of RaE across feature-based and vision-based, simulated robot locomotion and manipulation settings and standardised RL control benchmarks.

### 3.1 DOMAINS

We consider the following set of challenging domains (visualised in Figure 2):

**Locomotion Soccer** For locomotion, we consider a robot soccer task (Haarnoja et al., 2023) where a simulated OP3 humanoid robot (Robotis OP3, 2022) is rewarded for scoring goals (with additional shaping rewards) against an opponent. We consider two versions of this task: using proprioception and task-specific features (Locomotion Soccer State) and using proprioception with egocentric visual features (Locomotion Soccer Vision). This second variant is particularly challenging since the agent must learn to score from a sparse reward signal and visual features that make the environment partially observable. Further information on this domain can be found in Appendix A.1. We use MPO (Abdolmaleki et al., 2018) with a distributional critic (Bellemare et al., 2017; Barth-Maron et al., 2018) as underlying off-policy RL algorithm. For our experiments we gather training data of 4e5 and 2e5 episodes [2] for state and vision respectively.

**Manipulation RGB Stacking** For manipulation, we consider the task of stacking parameterized color-coded objects from Lee et al. (2021). This task builds on visual inputs and only sparse rewards. We use the multi-task SAC-Q off-policy RL algorithm (Riedmiller et al., 2018) in this domain. Further information can be found in Appendix A.2. The offline dataset here consists of 15e4 episodes of training data from a prior experiment with the same algorithm.

**RL Unplugged** Finally, we evaluate with the offline RL benchmark and dataset RL Unplugged (Gulcehre et al., 2020), which includes offline data on various simulated control domains. Within these, we focus on the three most challenging Control Suite domains (Tassa et al., 2020): humanoid run, manipulator insert peg and manipulator insert ball. The benchmark dataset contains

---

[1]We store and reuse all training data across experiments, not just trajectories left in the final replay.
[2]Throughout the text we use the scientific notation shorthand where 4e5 refers to $4 * 10^5$.

3000 episodes for humanoid run and 1500 episodes each for the manipulator domains. We use RaE with the offline RL algorithm CRR (Wang et al., 2020), which to our knowledge has state-of-the-art performance for pure offline learning on this benchmark.

The offline data in this setting is notably different from the other domains. In order to collect the data, the authors follow a sub-sampling criterion (see Gulcehre et al. (2020) for details) to select a subset of episodes which are then randomly stored as single-step trajectories. This is in contrast to our simple protocol of collecting and reusing all data across training with a single algorithm. We include this domain to illustrate the robustness and flexibility of RaE as well as to simplify future extensions and comparisons. More details on all domains are presented in Appendix A.

## 3.2 BASELINES

We compare RaE against a number of strong baselines to illustrate its effectiveness.

**Fine-tuning** We first train a policy and state-action value function offline using CRR (Wang et al., 2020). We evaluate a pure behavior cloning agent (BC) and two variants of CRR, CRR binary and CRR exp (see Appendix B). We then choose the best performing seed from across these runs and fine-tune (both policy and value function) online.

**AWAC** AWAC (Nair et al., 2020) starts by learning entirely offline for a fixed number of pre-training steps before switching to online iteration. The main difference to finetuning applies to the online phase, where it uses a offline/online shared replay buffer with a specific algorithmic formulation. The authors of this work prescribe 25e3 pre-training steps offline before incorporating online data. For our sweep we also considered 5e4 steps and 1e5 steps. We also sweep over values for the Lagrange multiplier $\lambda$ (0.3 and 1) as proposed by the original work.

**Random Weight Resetting** Reloading data for an experiment restart implicitly involves resetting network weights. Nikishin et al. (2022) observed that network resets on their own can be beneficial for learning. [3] To tease apart the effect of weight resets from the benefit of data reloading, we consider a baseline where all weights (policy, critic and optimizer) are reset every $K$ policy updates; where $K$ is set to the number of updates used to train the policy that generated the data. We sweep over reset frequencies of $K$, $K/10$ and $K/100$[4]. All experiments are run till convergence or at least $2 * K$ updates.

For each method, we report results of the best performing hyperparameter variant averaged across 5 seeds. Importantly, note that RaE did not require any tuning and we report results using the same hyperparameter set across all of the main results (50% offline data). While minor performance improvements with different parameters for RaE can be observed in extreme settings (see Table 1), we find RaE has very low sensitivity to hyperparameter choices across a range of diverse domains.

## 3.3 MAIN RESULTS

Figure 3 visualises our results on the Locomotion Soccer and Manipulation domains. It compares asymptotic performance achieved by RaE and other baselines after convergence as a bar plot with the dark solid line showing a 95% confidence interval around the mean averaged across 5 seeds after smoothing over 1000 episodes. As the figure shows, RaE consistently achieves the highest asymptotic performance across these tasks with a notable improvement on the challenging vision-based 'Locomotion Soccer Vision' domain. In the Manipulation settings, both finetuning and RaE perform similarly, although finetuning requires choosing the right algorithm and hyperparameter for offline learning.

Figure 4 compares RaE with all baselines that can use the RL Unplugged offline dataset. The figure shows the accumulated reward over the total number of additional *online* data consumed for each method. The performance of the best pure offline variant (CRR) is shown as a dotted blue line. We observe that RaE performs at-par or better than other comparable methods with the biggest difference being in the manipulator tasks. Methods that use offline learning (CRR and

---

[3]When using RaE, the data distribution throughout learning uses a fixed ratio between offline and online data, in addition to implicitly resetting network weights.

[4]Since the data collection methodology for the RL Unplugged domains is different and the number of updates unknown, we instead chose to sweep over values of 1e4, 1e5 and 1e6 updates for these domains

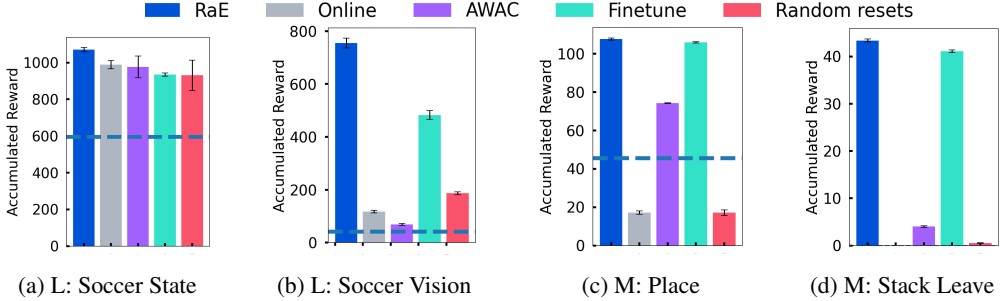

Figure 3: Comparison of RaE against baselines on the Locomotion Soccer and Manipulation domains (denoted by L: and M: respectively). Y-axis shows accumulated reward (undiscounted episode return) for each method; dotted blue line indicates offline learning performance with CRR. RaE consistently performs at-par or better than the baselines with a notable improvement on tasks involving vision.

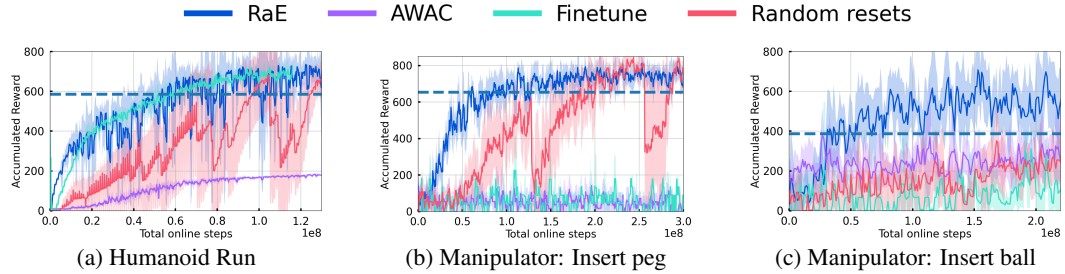

Figure 4: Performance of various mechanisms to reuse offline data using the RL Unplugged dataset. Accumulated rewards (undiscounted episode return) on the Y-axis are plotted against the total steps. Each curve plots the mean performance across 5 seeds with standard deviation shown as shaded regions. RaE consistently performs the best across all tasks.

finetuning) work well on the densely rewarded Humanoid run domain but peform poorly in the sparsely rewarded manipulator domains whereas RaE works well across all settings.

## 3.4 ABLATIONS

In this section, we present a set of experiments designed to answer the following questions:

- How much data is required for performance improvements?
- What kind of data is best for mixing: expert data, early training data or a mix?
- How sensitive is RaE to the ratio of online to offline data in different data regimes?
- Is there an advantage of applying RaE repeatedly across experiment iterations?
- Is RaE agnostic to the underlying choice of algorithm?

Unless otherwise specified, we present results on the 'Locomotion Soccer State' task with the DMPO algorithm.

**Analysis with varying data**    We answer the first three of these questions together in this section. For this analysis we use multiple subsets of the data collected for the 'Locomotion Soccer State' task, divided into three regimes:

- **High Return** We consider data generated only from the end of training from a complete training run. This corresponds to highly rewarding trajectories or 'expert' data.
- **Mixed Return** We consider data sampled uniformly at random throughout learning. This corresponds to a mixed regime with of high and low return trajectories.
- **Low Return** Finally, we consider data generated only from the start of training. This corresponds to early low return data.

| | High return | | Mixed return | | Low return | |
|---|---|---|---|---|---|---|
| Episodes Data mix | 10,000 | 100,000 | 10,000 | 100,000 | 10,000 | 100,000 |
| 50% Online | 51% | 90% | 80% | 119% | 97% | 112% |
| 70% Online | 80% | 110% | 101% | 124% | 108% | 121% |
| 80% Online | 98% | 118% | 106% | 126% | 110% | 123% |
| 90% Online | 108% | 120% | 113% | 120% | 108% | 121% |

Table 1: Performance comparison when using RaE in different data regimes with various mixes of online and offline data. Each cell represents asymptotic performance improvement as a percentage of the performance reached when learning from scratch *to convergence*(100% online). Colors are used to indicate the trend: red for below 90% performance, orange between 90 to 100% and blue when greater than 100%

High and Low Return data are sampled according to recency while Mixed return samples data uniformly. For each regime, we consider 2 datasets: one with 1e5 episodes[5] and another with just 1e4 episodes. The purpose of this analysis is to understand how best to adapt RaE in a regime with limited data. For each data subset we then consider data mixing at different ratios: in addition to the 50% mix that was used in the main results we consider mixes of 70, 80 and 90% online to offline data. For each setting, we consider the asymptotic performance reached when using RaE on the 'Locomotion Soccer State' task.

Table 1 shows the final asymptotic reward as a percentage of the *final* reward achieved when learning online from scratch, which corresponds to 4e5 episodes. To improve the ease of interpretation the cells are colored based on the reward achieved: yellow for below 90% performance, orange for between 90-100% and blue for greater than 100%.

A few interesting patterns emerge which we summarize below:

- In the lower data regime (1e4 episodes) a mixture with more online data is beneficial. However, as more data becomes available, a lower ratio works better (of 70-80% at 1e5 episodes). We hypothesize that using more online data for learning prevents over-fitting to a small set of offline trajectories.

- In the lower data regime, low return data tends to be the most beneficial. As the dataset size increases though, a mix of high and low return trajectories provide a greater benefit. Surprisingly, expert data is the least beneficial in both regimes. This indicates that the advantage of mixing data may stem from the benefit of having a larger state distribution with mixed rewards.

- Gains in performance can be achieved with as little as 1e4 episodes. This is particularly promising since even a small amount of prior data can considerably improve results.

**Iterative improvement** Figure 5a shows the improvement in performance when iteratively applying RaE on the 'Locomotion Soccer State' domain. For this experiment, we begin by generating a smaller dataset of 1e4 episodes and then apply RaE to that. We collect the data from this improved run (RaE iteration 1) and apply RaE again. We observe that there is a small gain in asymptotic performance and speed of learning even on the 2nd iteration although the performance plateaus on the 3rd iteration. This result suggests that in some settings it may be preferable to break a single training run into smaller runs for iterative performance improvements.

**Indifference to algorithm** For the main analysis of Section 3.3, we show that RaE can improve performance across a range of domains with different underlying algorithms. Figure 5b reinforces this point on the 'Locomotion Soccer State' domain where we show a considerable gain in performance using RaE with D4PG (Barth-Maron et al., 2018) for both data collection and mixing.

**Increased robustness** By applying RaE to combine data across seeds and hyperparameters from previous experiments we can add crucial robustness to the underlying agent's variance in performance. Figure 5c demonstrates robust performance when reloading data across random seeds in comparison to purely high or low performing data sources.

---

[5]The original dataset consists of 4e5 episodes.

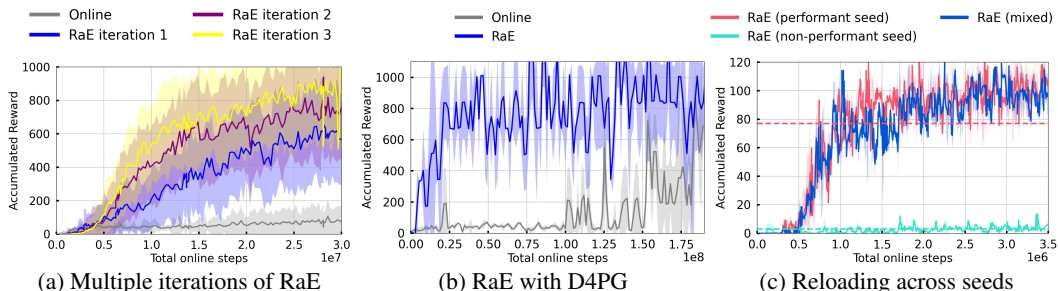

(a) Multiple iterations of RaE    (b) RaE with D4PG    (c) Reloading across seeds

Figure 5: a) RaE can be applied iteratively on the 'Locomotion Soccer State' domain. In some cases it may be preferable to have multiple smaller iterations although a performance plateau may be reached eventually. b) RaE on the 'Locomotion Soccer State' domain continues to improve performance with the D4PG algorithm. c) Combining data from seeds of a high variance experiment. RaE performs well even if just a single original seed achieves high reward. Performance of the original data generating experiments plotted as dotted lines. All figures plot mean accumulated reward against total online steps with standard deviation shown as a shaded region.

## 4    RELATED WORK

The most common paradigms to collect and reuse data in RL involve either offline agent datasets (e.g. RL Unplugged (Gulcehre et al., 2020) and D4RL (Fu et al., 2020)) or the use of human or expert demonstration trajectories (Vecerik et al., 2017; Bohez et al., 2022). While some work has focused on solving benchmark challenges (Wang et al., 2020; Kumar et al., 2020) others have focused on improving online performance by integrating offline data. For example, Singh et al. (2020a) and Ajay et al. (2021) learn hierarchical architectures offline and reuse them to solve more challenging tasks online. Related to our approach, Smith et al. (2023) mix experience using controllers designed for a different setting with online data to improve learning efficiency for quadruped robots. Our work instead highlights the merits of directly transferring data to improve asymptotic performance on a single domain.

**Learning from Data Combinations**    The idea of mixing expert data sources to bootstrap learning on the same domain has also been studied in a number of related works (Vecerik et al., 2017; Nair et al., 2018; Singh et al., 2020b; Walke et al., 2022; Davchev et al., 2021; Lee et al., 2022; Ball et al., 2023; Nair et al., 2020). While these methods have shown impressive results on a range of domains, the complexity and specificity of various algorithmic assumptions limit their generality. While Vecerik et al. (2017) also mix expert demonstration data into the replay, they include a prioritized replay and a mix of 1-step and N-step returns with L2 regularization in their setup. In a similar vein, Nair et al. (2018) introduce a BC-loss, Q-filter and state-reset mechanism as part of their data mixing approach. Other work relies on multi-stage procedures when mixing data: Singh et al. (2020b) distill prior data from skill experts to be later mixed with data from a scripted controller followed by the application of CQL; Walke et al. (2022) train a multi-task policy with separate forward and backward policies that are optimized separately. Even work that uses existing offline datasets as opposed to expert demonstrations inevitably include other components that increase their complexity. For instance, Lee et al. (2022) finetune an ensemble of policies trained offline using a prioritized replay and a density ratio to choose the data mixture to improve performance with the D4RL dataset. On the same domain, Ball et al. (2023) begin with a similar formulation to ours but then argue for random ensemble distillation and per-environment design choices and LayerNorm when training the Q-function with which they demonstrate improvements using a single off-policy algorithm. Finally, the AWAC algorithm (Nair et al., 2020) that we compare against sits in between finetuning and data mixing but uses a specific algorithmic formulation which introduces a number of domain-dependent hyperparameters.

In contrast, our focus is on the simplicity of the method: we empirically demonstrate the effectiveness and versatility of mixing previous data with a fixed ratio using many off-policy RL algorithms.

**Resetting Network Parameters in Reinforcement Learning**    A consequence of restarting experiments with mixed data is the resetting of neural network parameters. Resetting weights can change the learning dynamics and prevent overfitting which has been shown to be beneficial in both super-

vised learning (Ash & Adams, 2020) and RL (Nikishin et al., 2022; Schwarzer et al., 2023). Under certain settings where parts of a network are reset at specific intervals, these methods have been shown to greatly improve learning efficiency on challenging domains like Atari. In this work, we demonstrate that reusing previous experience can improve asymptotic performance on a range of tasks. Combined with its simplicity, we hope our approach can be included as part of the natural iteration cycle in RL and robotics in particular.

## 5 DISCUSSION

As RL continues to move from the object of study to a practical engineering tool for control (Degrave et al., 2022; Bellemare et al., 2020; Lazic et al., 2018; Lichtlé et al., 2022; Kaufmann et al., 2023; Osinski et al., 2020), simple and practical methods that can be generally applied will become increasingly important.

We can apply RaE throughout the lifetime of a project, across multiple experiments, algorithms, and hyperparameter settings. In the following paragraphs, we provide some intuition on some practical use-cases for RaE.

**Lifelong / Project-long learning** Research into novel domains nominally involves many iterations of trial-and-error to achieve state-of-the-art performance. While learnings from early iterations of experimentation inform algorithmic choices, the data collected in these trials is commonly discarded and never reused. Our initial analysis in Section 3.4 indicates that even low-return data can be useful to boost performance. With costs of data storage typically being far lower than compute, a different workflow where all experimental data (particularly in domains like robotics) is stored and reused to bootstrap learning could improve efficiency across project lifetimes.

**Multiple Source Experiments** RaE can also be applied in domains with many related source experiments. For example, consider many tasks defined via different reward functions but with the same underlying dynamics (e.g. the family of all manipulation tasks with a specific morphology). High-return data on some tasks may result in lower returns in another. However this data is still informative and may be useful in improving exploration when transferred.

**Multiple Hyperparameters and Seeds** Combining data from different variants of the same experiment (such as random seeds or hyperparameters) can enable better use of large, expensive sweeps. When reloading data with high performance variance over these parameters, the new experiment can benefit from the best option as initially discussed in Section 3.4.

**Potential Limitations and Strategies for Mitigation** Across all experiments, RaE demonstrates better or similar performance to state-of-the-art baselines for efficient off-policy RL without requiring per task hyperparameter tuning. However the applicability of the method is limited by the ability to reuse data. For example, changes in dynamics or experimental settings might invalidate previously collected data [6]. In such cases, an intermediate step to collect transitional data between old and new settings might be useful. Another challenge that may arise with widespread use of RaE is in the comparison of new algorithms. RL already suffers from issues of reproducibility (Henderson et al., 2018); if algorithms incorporate RaE, subtle changes in the data distribution when learning may create large differences in performance. One solution to this may be to specify deterministic orderings for benchmark datasets using fixed random seeds.

## 6 CONCLUSIONS

With this work, we introduce an effective and, importantly, simple workflow change for off-policy RL experimentation. Reloading data requires minimal infrastructure and can greatly improve performance as shown in Section 3. This makes it particularly useful from a project-long (or life-long) learning perspective in RL experimentation. We believe that as our understanding of RL improves and its use as an engineering and control tool becomes more commonplace, simplicity is key to effective integration.

---

[6]However in practice, this may not be much of an issue: see Appendix C.4

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

## A  ENVIRONMENT DETAILS

### A.1  LOCOMOTION SOCCER

The 'Locomotion Soccer' environment introduced by Haarnoja et al. (2023) consists of a Robotis OP3 (Robotis OP3, 2022) robot which is placed in a $4m \times 4m$ walled arena with a soccer ball. The robot must remain in its own half and scores goals when the ball enters a goal area $0.5m times 1m$ positioned across the center of the back wall in the other half. The robot must also contend with a random agent in the other half. The reward for this task is given by:

$$R_{goalscoring} = R_{score} + R_{upright} + R_{maxvelocity} \qquad (3)$$

where: $R_{score}$ is 1000 on the single timestep where the ball enters the goal region (and then becomes unavailable until the ball has bounced back to the robot's own half); $R_{maxvelocity}$ is the norm of the planar velocity of the robot's feet in the robot's forward direction; $R_{upright}$ is 1.0 when the robot's orientation is close to vertical, decaying to zero outside a margin of $12.5°$. Additionally, episodes are terminated with zero reward if the robot leaves its own half, or body parts other than the feet come within 4cm of the ground. The agent chooses a desired joint angle every 25 milliseconds (40Hz) based on input observations that include the joint angles, angular velocity and gravity direction of the torso. For 'Locomotion Soccer State' the agent also receives the coordinates of the ball, goal and opponent. For the 'Locomotion Soccer Vision' domain the robot instead receives a $40 \times 30$ render of the egocentric camera.

We use the experimental settings and networks described in Haarnoja et al. (2023) for this setting. We use a batch size of 128 with 5 seeds for the main results in Section 3 and a batch size of 256 with 2 seeds for the experiments in Section 3.4.

### A.2  ROBOT MANIPULATION

The RGB Stacking benchmark introduced by Lee et al. (2021) consists of parametric extruded shapes, which need to be stacked on top of each other by a fixed robotic manipulator, with the role of objects in the configuration being denoted via color coding. The benchmark uses a Rethink Sawyer robot arm, controlled in Cartesian velocity space on which a Robotiq 2F-85 parallel gripper is mounted.

We focus on the "skill mastery" challenge of the benchmark as described in Lee et al. (2021), where five fixed triplets of objects are used both for training and testing, with the goal being to place the red object on the blue one, while ignoring the green one. Each object triplet highlights a different manipulation aspect, such as balancing or reorientation.

For the SAC-X setup, we use a curriculum of sub-tasks that can be chosen by the scheduler. These follow the reward terms of the staged reward terms used in Lee et al. (2021), and intuitively contain sub-tasks for reaching, lifting, placing and stacking the red object. For evaluation, we report the final "stack-leave" reward term, which is a sparse reward given when the red object is placed precisely on the blue one, and the arm has been moved away by a predefined distance.

The observation provided to the agent consists of the images of three static cameras, as well as proprioception data from the robot and gripper, including joint angles, velocities and torques, as well as a simulated force-torque sensor attached to the wrist. It notably does not contain the tracked position of the objects, so the agent must learn to achieve the goal primarily from vision inputs.

In order to solve this task from visual inputs we use a network architecture where both policy and value function use a convolutional neural network with residual blocks (as used in (Espeholt et al., 2018)) that is then passed through an MLP for multi-task output for each of the subtasks. Both networks use (2, 2, 2) residual blocks with convolutional layers with 16, 32 and 32 channels respectively. The output of the policy network is used to condition a multi-headed Gaussian where each Gaussian output represents the policy output for a sub-task. Similarly the value function outputs a multi-headed categorical for each task . Appendix B describes how this can be used by the SAC-Q algorithm. All experiments are run with 5 seeds for the main results of Section 3.3 and 2 seeds for the ablations in Section 3.4.

### A.3 RL Unplugged

We consider 3 datasets from the RL Unplugged benchmark (Gulcehre et al., 2020): Humanoid run, Manipulator insert peg and Manipulator insert ball; which are categorized as 'hard' domains by Wang et al. (2020). The RL Unplugged datasets were collected on the DeepMind Control Suite (Tassa et al., 2018) implemented in the MuJoCo (Todorov et al., 2012) simulation framework.

Humanoid run consists of 3000 episodes of a 21 dimensional simulated humanoid walker that is rewarded for running forwards while staying upright. The data for this domain is generated using D4PG (Barth-Maron et al., 2018). The Manipulator insert ball and Manipulator insert peg domains consist of 1500 episodes each where a 5-dimensional manipulator is sparsely rewarded for inserting a ball and peg respectively into a specified hole. Data for this domain was generated using V-MPO (Song et al., 2020) since D4PG was unable to solve the tasks. The data generated on these tasks is then reduced in size via sub-sampling and the number of successful episodes in each dataset is reduced by $2/3$ to ensure the data does not contain too many successful trajectories. For all domains each episode consists of 1000 timesteps.

We use experimental settings as described in Wang et al. (2020) for this domain and use their networks for the offline and finetuning experiments. For the other baselines and RaE we use a an MLP architecture with sizes (256, 256, 128) for the policy and (512, 512, 256) for the value function. The policy network output is used to parameterize a Mixture of Gaussian (MoG) with 5 components as in Wang et al. (2020). Since the dataset only involves single-step transitions we use 1-step return for all baselines except the Random resets where we found 5-step returns performs significantly better.

## B Algorithm Details

In this section we describe the algorithms used in the main text in more detail.

**Policy Evaluation** We use the critic update described in Barth-Maron et al. (2018) for $N-$step returns:

$$(\mathcal{T}_\pi^N Q)(s_0, a_0) = r(s_0, a_0) + \mathbb{E}\left[\sum_{n=1}^{N-1} \gamma^n r(s_n, a_n) + \gamma^N Q(s_n, \pi(a_n|s_N))|s_0, a_0\right]$$

where the expectation is with respect to $N-$step transition dynamics and the distribution $Q$ and $\mathcal{T}$ represents the distributional Bellman operator. For our experiments we parameterize the critic as a categorical distribution with the number of bins set based on the domain. We set $N = 5$ for our experiments except in the RL unplugged domains where only single-step trajectories exist in the dataset. However, we use $N = 5$ for the random resets baseline in this domain since it can be run online and is not constrained by the limitation in the data.

**DMPO** For our online experiments in 'Locomotion Soccer' and 'Manipulation Stacking', we use the Maximum a-posteriori Policy Optimisation algorithm (MPO) (Abdolmaleki et al., 2018) to adhere to Haarnoja et al. (2023). MPO optimizes the RL objective in an E and M-step. The E-step update optimizes:

$$\max_q \int_s \mu(s) \int_a q(a|s) Q(s,a) \, da \, ds$$

$$s.\,t. \int_s \mu(s) D_{\mathrm{KL}}(q(a|s)||\pi(a|s)) \, ds < \epsilon$$

where $q(a|s)$ an improved non-parametric policy that is optimized for states $\mu(s)$ drawn from the replay buffer. The solution for $q$ is shown to be:

$$q(a|s) \propto \pi(a|s,\theta) \exp(\frac{Q_\theta(s,a)}{\eta^*}).$$

where $Q$ is computed as an expectation over the categorical distribution. In the M-step, an improved policy is the obtained via supervised learning:

$$\pi^{n+1} = \arg\max_{\pi_\theta} \sum_j^M \sum_i^N q_{ij} \log \pi_\theta(a_i|s_j).$$

$$s.\,t. D_{\mathrm{KL}}(\pi^n(a|s_j)||\pi_\theta(a|s_j))$$

**SAC-Q**  For the online experiments in 'Manipulation Stacking', we use the Scheduled Auxiliary Control (SAC-X) algorithm for multi-task exploration (Riedmiller et al., 2018). SAC-X builds on a multi-headed network architecture to represent both critic and policy, with a single "torso" per network shared across multiple tasks, and a separate output "head" for each task. At any given time, the active head to use is decided by a scheduler process. Here, the scheduler itself is also being learned, using the SAC-Q variant of the algorithm. At fixed intervals during each episode, a new sub-task may be chosen, and the scheduler trains a Q-function that optimizes the sequence of tasks executed such that it maximizes the total return of one task designated as the main goal.

To update the policy and Q-function, SAC-Q may use any update rule. Here, we use the MPO algorithm with a distributional critic, as described above.

**CRR**  For the offline experiments in Section 3.3 we use the Critic Regularized Regression (CRR) algorithm (Wang et al., 2020) which achieves state-of-the-art performance on the RL Unplugged benchmark. CRR uses the same policy evaluation step described above to learn a distributional critic. However the policy in CRR is trained by filtering data via the Q function by optimizing:

$$\arg\max_\pi \mathbb{E}_{(s,a)\sim D} \left[ f(Q, \pi, s, a) \log \pi(a|s) \right],$$

where the states and actions are drawn from the data source $D$ and $f$ is a non-negative, scalar function whose value increases monotonically with $Q$. We consider both variants of CRR introduced in the paper defined by choices of $f$:

$$f := \mathbb{1}\left[A(s,a) > 0\right], \tag{4}$$

$$f := \exp\left(A(s,a)/\beta\right), \tag{5}$$

where $A(s,a)$ is the advantage function which can be computed using:

$$A(s,a) = Q(s,a) - \frac{1}{m} \sum_{j=1}^m Q(s,a^j); a^j \sim \pi(.|s).$$

Following the authors of CRR, we refer to CRR with $f$ from Equation 4 as CRR binary and Equation 5 as CRR exp.

**AWAC**  For the AWAC baseline in the main text we follow the description from Nair et al. (2020) where learning proceeds in two stages: entirely offline initially and then with online data added to the same replay buffer. The transition between these two stages is defined by a hyper parameter which is set to 25,000 steps in the original work. For our experiments we sweep over an additional value of 50,000 steps and 100,000 steps for the RL Unplugged domain.

AWAC also introduces an algorithm which is similar to the CRR exp formulation described above with a slightly different notation that introduces a temperature $\lambda = \frac{1}{\beta}$ above. As per the original work we test the method with $\lambda$ set to 0.3 and 1.

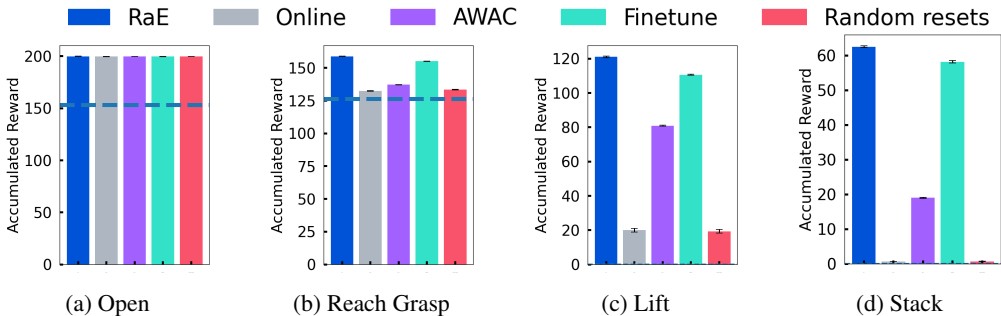

(a) Open      (b) Reach Grasp      (c) Lift      (d) Stack

Figure 6: Comparison of RaE against baselines on Open, Reach Grasp, Lift and Stack sub tasks in the Manipulation RGB stacking setting. Average performance across 5 seeds is plotted on the Y-axis with standard deviation as as dark lines.

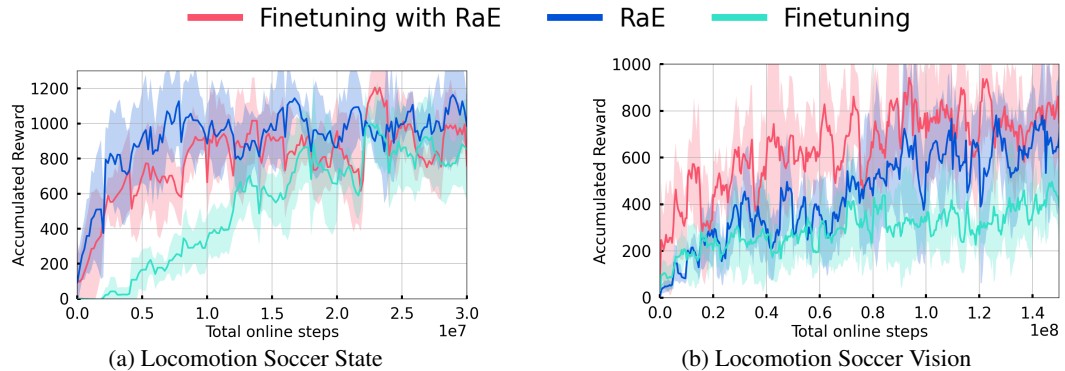

(a) Locomotion Soccer State      (b) Locomotion Soccer Vision

Figure 7: Comparison of finetuning, RaE and finetuning with RaE on the 'Locomotion Soccer' state and vision tasks (left and right respectively). Finetuning with RaE combines the best of both worlds: improved learning speed with high asymptotic performance. Accumulated reward on the Y-axis is plotted against total online steps on the X. Results are averaged across 5 seeds with the shaded region representing the standard deviation.

## C  ADDITIONAL RESULTS

### C.1  MANIPULATION RESULTS ON REMAINING TASKS

Figure 6 compares RaE against the other baselines from Section 3.3 on the remaining sub-tasks in the 'Manipulation RGB Stacking' setting. While all methods fare well on the easier tasks like 'Open' and 'Reach Grasp', RaE performs well on the harder 'Lift' and 'Stack' tasks similar to the trend on 'Place' and 'Stack Leave' that were presented in the main text.

### C.2  FINETUNING WITH RAE

The 'finetuning' baseline considered in the main experiments of Section 3.3 is orthogonal to the core idea of RaE and as such, we can apply RaE when finetuning. Figure 7 compares RaE , finetuning and finetuning with RaE on the 'Locomotion Soccer' tasks from state and vision. We observe that finetuning with RaE performs well across both domains matching the high asymptotic performance of RaE and learning faster by taking advantage of the network weights pre-trained offline.

### C.3  EFFECT OF MIXING RATIO ON FULL DATASET

The results presented in Table 1 show the effect of mixing different ratios of data when using RaE with smaller subsets of data (of 10,000 and 100,000 episodes). Figure 8a instead presents the same analysis when using the full dataset (of 4e5 episodes) on the 'Locomotion Soccer State' task. In this setting we see a clearer advantage of using offline data where a mixture of 50 to 70 percent of offline data works better than using more online data (90 %). In fact, as the figure shows,

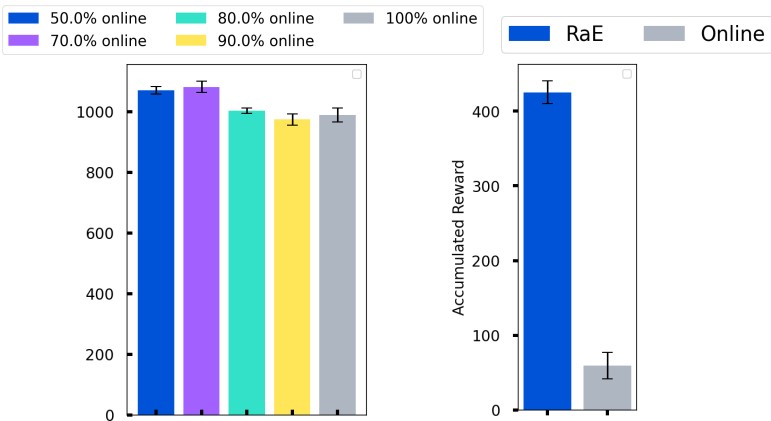

(a) Mixing ratio with full dataset (b) RaE with changing dynamics

Figure 8: (Left) Comparison with different mixtures of offline and online data when using RaE with 4e5 episodes of data on the 'Locomotion (State)' task. Accumulated reward (total undiscounted return) for each ratio of data is plotted. A mixture of 50 to 70 percent of online data works best. (Right) Comparison when using RaE to reload data to a new task with different dynamics. RaE is suprisingly robust and learns to perform better than learning from scratch indicating it's robustness to changes in dynamics.

a mixing ratio of 90 % may even slightly degrade performance when compared to learning purely online.

## C.4    EFFECT OF CHANGING DYNAMICS

In this section we analyze the effect of reusing RaE in a regime where the underlying dynamics change. We reuse the data collected in the 'Locomotion Soccer (State)' task but transfer to an environment where a mass is randomly attached to the left and right legs of the walker to perturb it's motion. Figure 8b shows that RaE continues to show an advantage as compared to learning from scratch even when the underlying dynamics of the task have changed. This somewhat surprising finding may be explained by modeling the environment as a partially observed MDP where under some conditions unknown to the agent, the dynamics of walking alter. Reusing data can still guide learning in such a setting showing the robustness of our approach.

