# OpenReview forum: "Replay across Experiments: A Natural Extension of Off-Policy RL"
_ICLR.cc/2024/Conference — ICLR 2024 poster_

### Official Review · Reviewer_gauC · 2023-10-29

**Soundness:** 3 good
**Presentation:** 3 good
**Contribution:** 4 excellent
**Rating:** 8
**Confidence:** 4

**Summary:**

The paper proposes Replay across Experiments (RaE) which is based on the simple concept of reusing experience from previous experiments to improve exploration and bootstrap learning. The proposed approach is employed with a number of existing algorithms in locomotion and manipulation tasks and shown to improve the learning efficiency.

**Strengths:**

The described approach is simple. It could be impactful as it is mostly algorithm-agnostic.

**Weaknesses:**

The main drawback is the lack of explicit explanations for the improvements observed. The limitations, perhaps such as increased storage memory could also be emphasized.

**Questions:**

1.	What motivated the reuse of previous experimental data? There is no explicit explanation for this.

2.	Do the age of trajectories matter? Eg: Suppose there is a limit on the data storage, would it be more valuable to add older trajectories to the data buffer or relatively newer ones?

3.	In the interest of comprehensiveness, I would have liked to see the benefits of this approach in Atari games as well. Does the approach improve the performance of say, DQN in Atari environments? Performance with methods like prioritized experience replay would also be interesting.

4.	The results in Table 1 are interesting. However, the reasons for the observed trends are not clearly explained.

5.	In Fig 1, the size of the ‘Data’ blocks is increasing, but is not immediately noticeable. It would be better to exaggerate the increase in block size to bring a reader’s attention to the data accumulation mechanism.

---

> ### Author Response · Authors · 2023-11-16
> **Response to reviewer gauC**
>
> Thank you for your helpful and positive feedback. We have updated the paper with suggested changes highlighted in red and are confident that the presentation is stronger thanks to them. We address specific questions below and have added a general comment for points shared across reviews. Please do not hesitate to ask for further clarifications.
>
> * What motivated the reuse of previous experimental data
>
> We demonstrate how reusing previous data improves performance across a range of diverse domains and algorithms. Related work has demonstrated a similar effect to some degree albeit with additional algorithmic complexities that render them difficult to implement. Our focus in this work is on analyzing an underlying simple core idea more directly. With the increasing proliferation of offline datasets for robotics domains, RaE is an effective tool that can be incorporated into existing algorithms and fits naturally into existing RL experimentation workflows.
>
> * “..would it be more valuable to add older trajectories to the data buffer or relatively newer ones?”
>
> The results of Table 1 indicate that the best way to use RaE with limited data is to combine trajectories sampled randomly throughout training (Mixed return) with a higher ratio of online to offline data (70-80 % works quite well). We have emphasized this point in the text.
>
> * Benefits of RaE on Atari games
>
> The `Random resets’ baseline we consider in our work is closely related to the work of Nikishin et al. [1]. They demonstrate an improvement of performance on Atari domains albeit with different underlying algorithms. We demonstrate RaE outperforms random resets across a range of diverse environments and algorithms with continuous action spaces; a finding which is likely to carry over to the Atari domain.
>
> * Reasons for results in Table 1.
>
> Table 1 presents an analysis of choosing different mixing ratios when running RaE with smaller datasets (10,000 and 100,000 episodes). Importantly, for all the main results shown in Figure 3 and 4 a fixed 50-50 ratio works well across domains. However, when the amount of data is restricted, using more online data is beneficial and prevents overfitting to a narrow distribution of offline trajectories. We have clarified this point in the description of Table 1.
>
> * “..exaggerate the increase in block size to bring a reader’s attention to the data accumulation mechanism.”
>
> Thank you for the suggestion. We have updated the paper with a new version of the figure that highlights the increasing block size of the data on each iteration. Please let us know if further improvements can be made.
>
> [1] Nikishin, Evgenii, et al. "The primacy bias in deep reinforcement learning." International conference on machine learning. PMLR, 2022.

---

> > ### Comment · Reviewer_gauC · 2023-11-20
> > **Thanks**
> >
> > Thanks for your responses.
> >
> > Regarding the reuse of old/new trajectories - As the authors pointed out, using a mixed strategy works well. However, mixed return samples data uniformly. My question was - how would the performance be affected if this sampling was not done uniformly, but proportional (or inversely proportional) to recency? Perhaps this is an additional set of ablations that the authors could consider including.
> >
> > Regarding Atari experiments - Although RaE shows superior performance over the `Random resets' baseline (which shows relatively improved performances in Atari), it would be better to show RaE's actual performances in Atari, as the attributes of that environment (with image-based observations and discrete actions) are fundamentally distinct from the ones shown in the current paper. However, I understand if the experiments cannot be added at this stage, given the limited time frame.
> >
> > Regarding my point on bringing attention to the data accumulation mechanism, my suggestion was to increase the sizes of the second and third `Data' blocks relative to the first one, but I leave this choice to the authors.

---

> > > ### Author Response · Authors · 2023-11-20
> > > **Response to Reviewer gauC**
> > >
> > > Thank you for your useful comments and for the quick response in engaging with the review process. We have made further changes to the paper and have addressed your comments below. We are happy to continue to discuss improvements to our work.
> > >
> > > * “How would the performance be affected if this sampling was not done uniformly? How would the performance be affected if this sampling was not done uniformly, but proportional (or inversely proportional) to recency? ”
> > >
> > > Table 1 shows the performance with subsets of data across three regimes: High return, Mixed return and Low return. For the Mixed return regime data is sampled uniformly across the training dataset. However, for the Low and High return regimes data is sampled according to recency. Low return corresponds to the data from the beginning of training and High return corresponds to the data from the end of training from a fully converged training run. Therefore, these results currently indicate how performance is affected when sampling according to recency: we find that in general Low return data is better than using `expert` or High return data. We have updated the description of these data regimes in the main text to clarify this point.
> > >
> > > * “Atari experiments”
> > >
> > > Thank you for your response. We agree that Atari environments may have attributes like discrete action spaces that are distinct from the ones currently shown in the paper. Note that the `Locomotion (Soccer) Vision` and `Manipulation` environments discussed already include image observations from first and third person viewpoints. We are working towards including the Atari environments which we can include in the final version of the paper if accepted.
> > >
> > > * “Data blocks sizes”
> > >
> > > We increased the font size and weighting of the `Data` blocks to highlight the growing dataset sizes. We have now updated the figure to further increase the size of the data blocks and hope this latest version fully addresses the reviewer’s concerns.

---

> > > > ### Comment · Reviewer_gauC · 2023-11-22
> > > >
> > > > Thanks for your responses and clarifications. You are right - some of the environments are indeed image-based. Apologies for my oversight.

---

### Official Review · Reviewer_u3fY · 2023-10-31

**Soundness:** 3 good
**Presentation:** 3 good
**Contribution:** 2 fair
**Rating:** 6
**Confidence:** 4

**Summary:**

This paper presents a method to reuse experience from prior RL experiments and shows benefits across a variety of RL algorithms and experiments.

**Strengths:**

- The authors present a very simple idea that leads to improved performance across a variety of environments.
- The proposed method works well even with a small amount of prior data.
- The proposed method works well even with low return offline data, which makes the method much more useful in practice.

**Weaknesses:**

- It is not clear to me if "Total online steps" in the figures includes the steps from prior experiments or not, so I'm concerned about the fairness of the comparison between RaE and baselines. If the authors can clarify this point then I may be willing to raise my score.

**Questions:**

- I'm a bit confused about the difference between Random Weight Resetting and RaE. The authors write that "Reloading data for an experiment restart implicitly involves resetting network weights", which makes me think that RaE and Random Weight Resetting are very similar. However, this clearly isn't the case since RaE performs better. Can the authors clarify the difference?
- Across all figures does "Total online steps" include the steps from prior experiments for RaE? If not, I'm concerned that it may not be a fair comparison.
- The authors write, "At the beginning of each training run, policy and value-function are re-initialized in line with stand-alone experiments". I'm curious if re-initializing vs not re-initializing at the start of each training run makes a difference in performance?
- Under "Potential Limitations and Strategies for Mitigation" the authors write that "changes in dynamics or experimental settings might invalidate previously collected data." Have the authors actually tried experimenting with changing dynamics across training runs. I'd be curious to see how much of a negative effect this would actually have in practice.

---

> ### Author Response · Authors · 2023-11-16
> **Response to reviewer u3fY**
>
> Thank you for your helpful and constructive feedback. We have updated the paper with suggested changes highlighted in red and are confident that the presentation is stronger thanks to them. We address specific questions below and have added a general comment for points shared across reviews. Please do not hesitate to ask for further clarifications.
>
> * Total online steps
>
> We have updated Figure 4 to include the data from the offline dataset. The relative performance of RaE against the baselines remains unchanged and importantly, RaE continues to show improved asymptotic performance on challenging domains.  We hope this addresses your original comment and will help improve your original score. If there are further questions, please let us know.
>
> * Difference between Random resets and RaE
>
> When restarting an experiment with RaE, in addition to resetting the weights of the network, the data distribution throughout learning utilizes a fixed ratio of mixing between offline and online data. In contrast, `Random resets’ only reload network weights and then use data from the replay buffer for learning in which older data is overwritten with time. In addition, RaE is more practical to implement since a period to reset does not have to be chosen and fixed a-priori; experiments can be restarted and data mixed naturally during the course of experimentation with repeated iteration.
>
> * “I'm curious if re-initializing vs not re-initializing at the start of each training run makes a difference in performance?
>
> This is a very good point. The choice of re-initializing weights is orthogonal to RaE and can be incorporated on top of RaE. Figure 7 of Appendix C.2 highlights this by comparing the performance of RaE with and without reloading weights on the `Locomotion’ domains. We find reinitializing weights can improve learning speed without hurting asymptotic performance.
>
> * “Have the authors actually tried experimenting with changing dynamics across training runs.”
>
> Our comment regarding changing dynamics alludes to a theoretical limitation where an extreme change in the dynamics could render the transitions in the offline data invalid. However, as you note, this may not be much of an issue in practice. To test this, we conducted an experiment presented in Appendix C.4 where we transfer data from the `Locomotion Soccer (State)’ task to a new setting where a mass (of 300g) is randomly attached to the legs of the walker to perturb its motion. Surprisingly we find RaE works well in practice in this setting and can learn a partial solution to the task while learning online from scratch fails to learn any solution.

---

> > ### Author Response · Authors · 2023-11-22
> > **End of rebuttal**
> >
> > Thank you once again for the valuable feedback. As a quick summary, based on your comments, we have updated Figure 4 to include total online steps and have included new results with changing dynamics in Appendix C. With the discussion period coming to an end soon, please let us know if there are any other points we can clarify in addition to our responses above.  If not, we would be grateful if you could consider increasing your score based on our feedback.

---

> > > ### Comment · Reviewer_u3fY · 2023-11-23
> > >
> > > Thanks for the clarifications! My main concern has been addressed, so I am raising my score.

---

### Official Review · Reviewer_fgij · 2023-10-31

**Soundness:** 3 good
**Presentation:** 3 good
**Contribution:** 2 fair
**Rating:** 6
**Confidence:** 4

**Summary:**

The authors propose Replay across Experiments (RaE), in which all past experiment data is stored in the replay memory for training. In essence, data is never discarded. Data from all past experiment trials is stored and reused, not just the trajectories left in the buffer at the end of the previous experiment. The authors recommend a default mixture of 50-50 offline/online data (where “online” data is from the current experiment) to obtain good performance without tuning this hyperparameter. The authors compare RaE against other common strategies (fine-tuning, AWAC, and parameter resetting) combined with algorithms such as DMPO and D4PG in control benchmarks including Locomotion Soccer, Manipulation RGB Stacking, and RL Unplugged.

***Rebuttal: score raised from 5 to 6**

**Strengths:**

- The method is simple to implement compared to many existing data-reuse RL methods.
- The experiments in the paper are comprehensive, testing a multitude of diverse methods in a number of high-dimensional control environments. Several strong baselines from the literature are compared against. In spite of this simplicity, RaE can achieve strong performance in these tasks. The breadth of the results demonstrate the generality of RaE.
- The paper is well organized and includes nice discussions for related work and practical use cases.

**Weaknesses:**

- The main insight of the work, while interesting, is a small contribution. As the authors note in the background section, other methods already use data stored from previous experiments, so the only novelty here is that *data is not discarded*. There is no theoretical analysis in the paper. Without significant novelty or theory, the paper depends solely on its empirical results.
- Storing all previous data is memory intensive, which is why offline RL generally uses more complicated techniques to learn from limited data. The authors do not discuss this drawback of their method, but I could see it being a bottleneck in long experiments with high-dimensional observations.
- It is unclear that replaying offline data from a long time ago is as beneficial as the authors claim. Table 1 seems to indicate that performance improves almost monotonically as the proportion of online data increases to 90%. The authors claim that “as more data becomes available, a lower ratio [of 70-80%] works better,” but this only happens occasionally, and the performance improvement is small (about 2-5%). Without any measure of significance provided, it does not seem that more than 10% offline data actually helps much.
- The results are rather noisy and would benefit from increasing the number of trials (which is currently only 5). I would recommend that the authors use a 95% confidence interval instead of standard deviation and apply a 100-episode moving average (if they are not already doing so already) to make the results easier to read. Currently, some of the standard deviations are overlapping, but I think the results would be significant if confidence intervals are used instead.

**Minor edits:**
- In-text citations are not rendered correctly on the last line of page 1.
- In the background section, the range of discount values should be mentioned: $\gamma \in [0,1]$.
- I think it would be helpful to the reader to define the scientific notation shorthand the first time it is used, e.g., $\text{4e5} = 4 \times 10^{-5}$.

**Questions:**

1. What is meant by “Accumulated Reward” in the y-axes of Figures 3, 4? Is this the undiscounted episode return?
1. What is meant by “from scratch to convergence” in Table 1’s caption? Does that mean that exactly 100% refers to the final performance obtained by a pure online-data agent?

---

> ### Author Response · Authors · 2023-11-16
> **Response to reviewer fgij**
>
> Thank you for your helpful and constructive feedback. We have updated the paper with suggested changes highlighted in red and are confident that the presentation is stronger thanks to them. We address specific questions below and have added a general comment for points shared across reviews. Please do not hesitate to ask for further clarifications.
>
> * “The main insight of the work, while interesting, is a small contribution.”
>
> We respectfully disagree on the scale of contribution of our work. As you note, we present a highly-effective, simple method that can easily be introduced to RL methods and provide a comprehensive evaluation showing good performance compared to several strong baselines across a breadth of domains. As we discuss in the main text, other works that use a similar idea do so in an incidental manner, and commonly focus on additional algorithmic changes that add considerable complexity. By focusing on the core approach and combining its simplicity with a strong empirical contribution we believe our work can translate to practical real world impact across a range of RL domains.
>
> * “Storing all previous data is memory intensive”
>
> We agree that memory is not free, but as Reviewer wVcc points out, and as witnessed by the increasing availability of large datasets in RL and robotics [1], memory is relatively cheap. Large, shared datasets for robotics are becoming more widely available and are likely to be integrated into existing RL workflows. As we show, RaE improves asymptotic performance on many domains: in many cases it may be cheaper to scale up data storage rather than compute to achieve a similar final performance. And as illustrated by Table 1, we can achieve strong results by storing relatively small subsets of data randomly throughout training.
>
> * Confidence intervals instead of standard deviations.
>
> We agree that the visual presentation of data is important.  We have replaced the error bars for the plots in Figure 3 (and Figure 6 in the Appendix) to instead show a 95% confidence interval over the mean after smoothing over 1000 episodes and have updated the description in Section 3.3 to reflect this.
>
> * 90% online data works best in Table 1
>
> The results in Table 1 discuss how best to adapt RaE to regimes where data is sparse. As you point out, using more online data is a reasonable strategy in this setting. However note that mixing some offline data is preferable to learning purely offline and all the results in Figure 3 and 4 show an improvement in asymptotic performance with a simple 50/50 ratio of mixing data.
> Additionally, we have updated the paper to include an additional result in the Appendix C.3 that illustrates the effect of mixing different data ratios when using the full dataset (of 4e5 episodes). This result represents many project settings and shows that when more data is available, a mixing ratio of 50 to 70 percent of online data works better than a mix with too much online data.
>
> * Accumulated Reward
>
> Yes, by accumulated reward we mean the undiscounted episode return achieved by the agent. We have updated the captions of Figures 3 and 4 to reflect this.
>
> * Table 1 caption ‘from scratch to convergence’
>
> The results in Table 1 present the performance of RaE with small subsets of data as a percentage of learning from scratch. By ‘from scratch to convergence’, we want to highlight that the agent performance being compared against is not at the 10,000 or 100,000 episode mark but with an online agent trained until it converges.
>
> * Minor edits
>
> Thank you for pointing these out to us. We have made corrections to the text.
>
>
> [1] Open X-Embodiment: Robotic learning datasets and RT-X models. https://robotics-transformer-x.github.io.

---

> > ### Comment · Reviewer_fgij · 2023-11-18
> >
> > Thanks for making the suggested paper updates. I do agree that the simplicity of the approach is valuable, and I also see that argument that memory is cheaper than compute. The method is very general and can be combined with most (off-policy) algorithms, so I think it can be impactful.
> >
> > Given that the paper does not advance any new theories or extremely new ideas, though, I think it is really important to demonstrate beyond reasonable doubt that RaE is effective in practice. The current empirical evaluation is already quite extensive and I think there is a lot of potential in this regard. However, there are a lot of theoretical issues with off-policy training, like the effect of mixing stale data distributions on the fixed points of the function approximation, that worry me that the observed results will not generalize. See, for example, Figure 5 of [1] which shows the state weighting greatly impacts the solution quality, and the on-policy weighting usually leads to lower approximation error.
> >
> > What would help convince me is a clearer ablation study. Based on the results in Table 1 / Figure 8a, it is still not clear to me that mixing offline data is always helpful against pure online agent. This may stem from a misunderstanding on my part, but why is a 100% online-data mix not compared in this study, or is it? Is that what is meant by the 100% score relative to "from scratch to convergence"?
> >
> > Would it also be possible to show learning curves instead of summarized final performance (hopefully this isn't too hard if you still have the data)? In my opinion, the most interesting case is the mixed-data regime, since that is used by your main experiments. It also concerns me that the datasets used in these ablations are smaller than the ones used in your main experiment. In off-policy RL, it is not always the case that more data is better, due to the reasons I mentioned above. For example, see this paper that showed that a DQN-like agent can actually perform worse as the memory size increases [2]. I think the ablation would be much stronger if it closely matches your main experiment setup and clearly demonstrates that a variety of data mixtures are better than a pure online agent, throughout training and not just the final performance.
> >
> > [1] A Generalized Projected Bellman Error for Off-policy Value Estimation in Reinforcement Learning. Patterson, White, and White, 2022.
> > https://arxiv.org/pdf/2104.13844.pdf
> >
> > [1] A Deeper Look at Experience Replay. Zhang and Sutton, 2017.
> > https://arxiv.org/pdf/1712.01275.pdf

---

> > > ### Author Response · Authors · 2023-11-20
> > > **Response to Reviewer fgij**
> > >
> > > Thank you for your useful comments and quick response in engaging with the review process. We have made further changes to the paper and have addressed your comments below. We hope that you will consider increasing your score and are happy to continue to discuss improvements to our work.
> > >
> > > * “It is not clear to me that mixing offline data is always helpful against pure online agent”
> > >
> > > We show that across a range of diverse environments and algorithms a 50-50 ratio of offline and online mixing improves asymptotic performance when simply reloading data from a previously completed run. We would like to clarify though that we do not claim that mixing offline data is always helpful against a pure online agent. In fact as the ablation in Table 1 shows, with very little data (10,000 episodes), a 50-50 mixing ratio works less well than a purely online agent. In such cases, we find a higher ratio of online data can improve performance over the online run. The performances in Table 1 are represented as a percentage compared to training entirely online in that setup.
> > >
> > > * Concerned that datasets used in ablations are smaller than in the main experiment: ablation would be stronger if it closely matches the main experiment setup. Why is a 100% online-data mix not compared in this study?
> > >
> > > Thanks for the comment. The follow-up ablation in Figure 8a shows the effect of data mixing with the exact experimental setup as described in the main experiments. We have updated this figure and corresponding text to include the 100% online performance as well as per your suggestion.
> > >
> > > In addition, we are also running the ablation in Figure 8a on the `Locomotion (Soccer) Vision` task.  If space allows, we will try to incorporate this into the main body of the paper. These results may not be ready before the end of the review process but we will try our best to update them in time.
> > >
> > > The purpose of the original ablation was to investigate how RaE performs with different data mixing ratios under various regimes where less data is available. We chose to report this as a percentage of the performance achieved when training entirely online for ease of understanding. To clarify by 100% score relative to `from scratch to convergence’ we mean an agent trained entirely online (i.e. a 100% online-data mix) with the same experimental setup as the other methods in the ablation, until it converges. With these new results and clarifications, we hope that you will consider increasing your score and are happy to discuss further.

---

> > > > ### Comment · Reviewer_fgij · 2023-11-23
> > > >
> > > > Thank you again for the clarifications. I appreciate the new ablation results in Figure 8a. The comparison to 100% in the bar chart makes the performance improvement much clearer.
> > > >
> > > > Similarly, would it be possible to add another row to Table 1 to show the 100% online performance in the same manner? This would obviously just be a row of 100%'s, which may seem redundant, but I think it would make the table's purpose clearer. Part of my initial skepticism of the method was because it appeared the 100% case was being intentionally ignored. The 100% row's values could be indicated in a different color (e.g., black) to make it very clear that it is the baseline. The caption could also updated to emphasize that the agent trained "from scratch to convergence" is a 100% online-data mix. All of these small changes would make the figure more readable in my opinion.
> > > >
> > > > I also think it would be great to briefly discuss some of the theoretical issues related to off-policy learning that I talked about above. The empirical results show that these are not too much of an issue in practice, but I think it is important to mention them.
> > > >
> > > > In the meantime, I appreciate the improvements made to the paper. I have raised my score.

---

### Official Review · Reviewer_wVcc · 2023-11-06

**Soundness:** 3 good
**Presentation:** 4 excellent
**Contribution:** 2 fair
**Rating:** 6
**Confidence:** 4

**Summary:**

The paper discusses a new setting, where data is shared across experimental runs to improve the performance of the policy. A simple strategy, adding data from previous experiments to the initial replay buffer, is explored and shown to be an effective approach. Various factors are investigated including the quality of the data used and the amount of data kept. These experiments are done with a variety of policy optimizatino algorithms and robotics benchmarks.

**Strengths:**

- The approach of keeping data from previous experiments seems to be very relevant in practical scenarios where our main goal is to train an agent with strong performance. In that case, given the cheap cost of memory, it would be sensible to keep data from previous runs for the benefit of future experiments.
This seems like an understudied topic and it's great that this paper discusses it.

- There's a nice variety of environments that are used, including some more complex ones with a good mix of algorithms too.

- The writing and organization is clear, making the paper easy to read. The paper has a distinct focus which helps gets the message across too.

**Weaknesses:**

- In the current paper, most of the experiments run the same learning agent on the same environment with the RaE algorithm but the method is pitched as being helpful for boosting learning between different experiments with potential differences in experimental conditions.
See Questions.

- There are other simple algorithms for this across-experiment setting that would be interesting to investigate. See Questions.

**Questions:**

- Aside from RaE, another natural baseline for this across-experiment setting would be to use behaviour cloning (or distillation) on the previously trained agents. Have you considered doing so?

- RaE only incorporates the previous data at the beginning of optimization. What about training on the data throughout the optimization process? For example, keeping the buffer of previous data and sampling from it occasionally or mixing in samples into minibatches.

- As I mentioned in strengths section, I think training across-experiments could be relevant for practical purposes. How do you see it being used in a scientific context? It could be difficult to fairly assess algorithms if the data they have access to depends on the sequence of previous experiments done. i.e. an advantage of discarding previous data is that algorithms start on equal ground in different papers, allowing fair comparisons.

- I'm a bit surprised that doing some offline learning first is so detrimental to the policy (Fig. 4, finetuning and AWAC). Intuitively, I would guess that there should be a jumpstart in the performance due to the additional offline training. Could you clarify this?
Do you have any hypotheses why these methods don't perform very well here?

- Currently, the experiments that use RaE are quite similar to simply using resets but at the end of each experimental run. It would be interesting to see some experiments where the same algorithm was not used in consecutive experiments or, at least, changing hyperparameters. This could better simulate the development process of an RL algorithm.
As a suggestion, I would be curious to see if hyperparameter optimization could be made easier. Since RL agents can be sometimes hard to run initially on a new problem, we might be able to more easily identify which hyperparameter settings are promising by giving the agent additional data.

---

> ### Author Response · Authors · 2023-11-16
> **Response to Reviewer wVcc**
>
> Thank you for your helpful and constructive feedback. We have updated the paper with suggested changes highlighted in red and are confident that the presentation is stronger thanks to them. We address specific questions below and have added a general comment for points shared across reviews. Please do not hesitate to ask for further clarifications.
>
> * “What about training on the data throughout the optimization process?”
>
> RaE mixes online data and offline data in a 50/50 ratio for each minibatch sampled throughout training (and not just at the beginning). We have updated the text (page 3) to clarify this.
>
> * RaE in the context of the scientific method for reproducible and comparable results.
>
> This is a very good point; thank you for highlighting it. Reproducibility in RL is already a challenging open problem [1]. If RaE is more widely adopted, it may add another dimension to this issue: subtle changes in the ordering of data when mixing may lead to changes in final performance. One solution to this could be to standardize and open source offline data sources for benchmark domains (e.g. RL Unplugged) and define deterministic orderings when reading from the data using, for example, fixed random seeds. We have included a brief discussion on this point in the Discussion section (Section 5).
>
>
> * “Why does offline training work so poorly?”
>
> In Figure 4, fine-tuning and AWAC learn on the Humanoid Run domain but perform poorly  on the Manipulator domains of RL Unplugged. Humanoid run uses a dense reward signal that incentivizes the agent for moving forward. In contrast, the Manipulator domains provide sparse rewards that are triggered in stages when a robotic arm moves to a peg (or ball) and inserts it into a hole. Learning offline from limited data in sparse reward settings is particularly challenging which is why we see a strong benefit of using RaE to mix in online data to improve asymptotic performance.  Additionally offline learning may also overfit to the limited set of trajectories in the dataset, making further improvement through fine-tuning difficult.We have added a sentence to Section 3.3 to highlight this.
>
> * “Additional baseline: BC and distillation to previous agent.”
>
> When fine-tuning, we consider two baseline agents trained with the offline data: one that uses CRR and another that uses BC. We then select the best performing variant among these to finetune. We believe this addresses your point but are happy to clarify and run another baseline (time permitting) if required.
>
> * “.. more easily identify which hyperparameter settings are promising by giving the agent additional data.”
>
> Yes, the faster iteration time when using RaE could be useful to search for better hyperparameters to use in new settings from scratch. However, it is important to note that hyperparameters that are tuned in a setting that reuses data may not directly transfer over to settings where data is not reused. For example, learning rates and optimizer parameters may not be directly transferable.
>
>
>
> [1] Henderson, P., Islam, R., Bachman, P., Pineau, J., Precup, D., & Meger, D. (2018). Deep Reinforcement Learning That Matters. Proceedings of the AAAI Conference on Artificial Intelligence, 32(1). https://doi.org/10.1609/aaai.v32i1.11694

---

> > ### Author Response · Authors · 2023-11-22
> > **End of rebuttal period**
> >
> > Thank you once again for the valuable feedback. As a quick summary, based on your comments, we have updated the limitations of offline learning in Section 3.3, added a discussion on RaE and reproducibility in Section 5 and have clarified the description of the algorithm in page 3. With the discussion period coming to an end soon, please let us know if there are any other points we can clarify in addition to our responses above. If not, we would be grateful if you could consider increasing your score based on our feedback.

---

> > ### Comment · Reviewer_wVcc · 2023-11-22
> >
> > Thank you for the clarifications. I had misunderstood the RaE algorithm slightly. It maybe nice to emphasize this difference between RaE and random resets as done in the response to reviewer gauC "When restarting an experiment with RaE, in addition to resetting the weights of the network, the data distribution throughout learning utilizes a fixed ratio of mixing between offline and online data."  Based on the responses to the other reviewers and this one, I'd be willing to increase the score.

---

### Author Response · Authors · 2023-11-16
**General comments to all reviewers**

Thank you for taking the time to provide thorough feedback. We hope to address all open questions and improve the paper based on your perspectives. A couple of points were shared across reviews and we will address them here. We furthermore add specific details in the responses to each review.

1. Contribution of our work

Several reviewers provided a medium to low Contribution score for our work. At the same time, all reviewers were positive about the strengths of our method:
  *  ‘This seems like an understudied topic and it's great that this paper discusses it’ (Reviewer wVcc);
  * ‘experiments are comprehensive; can achieve strong performance against several strong baselines’ (Reviewer fgij);
  * ‘a very simple idea that leads to improved performance across a variety of environments.’ (Reviewer u3fY) and
  * ‘..impactful as it is mostly algorithm-agnostic.’ (Reviewer gauC).

We agree with the reviewers that the strength of our work is in presenting an easy to implement, algorithmic-agnostic approach that shows strong performance across a range of diverse domains and algorithms. The focus of our work was to underscore the simplicity of the method that uses existing concepts in off-policy RL with a thorough empirical validation. We believe this is important as it highlights that our method can naturally be integrated into the workflow of RL practitioners and can thus have significant real-world impact. We believe publishing simple, but impactful ideas remains a challenge in our field and is something that we as a community should work on improving.

2. Changes to paper

We have updated the paper with changes to the text based on individual reviewer feedback and have updated the plots based on the feedback from Reviewer fgij and u3fY. All the changes are highlighted in red for convenience. We believe the paper is much stronger thanks to the constructive feedback and would once again like to thank all the reviewers for engaging with the review process.

---

> ### Comment · Reviewer_gauC · 2023-11-20
> **Inaccurate rephrasing**
>
> I noticed that Reviewer wVcc's comments are slightly misrepresented here. The reviewer's comment was that it is great that this paper studied an understudied topic, and not that this is a great paper that discusses an understudied topic. It is just a subtle difference in wording, but I believe it misrepresents the original comment. I assume this was unintentional, but I just wanted to point it out to give the authors an opportunity to correct it.

---

> > ### Author Response · Authors · 2023-11-20
> > **Response to rephrasing**
> >
> > We sincerely apologize: the mistake was unintentional. We have edited and corrected the original response to reflect Reviewer wVcc’s view verbatim: “This seems like an understudied topic and it's great that this paper discusses it.”. Our aim with these comments was to highlight the positive feedback from the reviewers regarding the scope of the problem and strength of our extensive empirical evaluation which we believe still holds true. However, once again, we sincerely apologize for the error on our part.

---

### Author Response · Authors · 2023-11-21
**End of rebuttal period approaching**

We would like to once again thank all the reviewers for taking the time to provide valuable feedback. The presentation and analysis in the paper is stronger thanks to this. As the rebuttal period is coming to a close soon, please let us know if there are any lingering questions or comments that we can address. We are in the process of running a final set of ablations and analysis based on your comments (see individual responses) but may not be able to update the paper before the end of the discussion phase.

We hope you will consider increasing your score and thank you again for your valuable time and contribution.

---

### Author Response · Authors · 2023-11-23
**Thanks to all reviewers**

Thanks again to all the reviewers for your constructive feedback and for the continuous engagement during the review process! We greatly appreciate the comments and discussion and have incorporated the feedback into the latest version of the paper. We think the work is much stronger thanks to this discussion.

---

### Meta-Review · Area_Chair_Wi1Y · 2023-12-24

**Metareview:**

### Summary
The paper proposes Replay across Experiments (RaE) which is based on the simple concept of reusing experience from previous experiments to improve exploration and bootstrap learning. The proposed approach is employed with a number of existing algorithms in locomotion and manipulation tasks and shown to improve the learning efficiency.


###  Strengths
+ algorithm-agnostic approach
+ strong empirical on a range of data regimes

### Weaknesses:
- lack of clear discussion on limitations such as increased memory footprint.

**Justification For Why Not Higher Score:**

The paper draft needs improvements to clarify the reviewer questions and additional experiments.

**Justification For Why Not Lower Score:**

By leveraging a simple yet effective the authors achieve strong empirical performance. The reviewers agree with the value of this method being valuable to the community.

---

### Decision · Program_Chairs · 2024-01-16

Accept (poster)